# Enzymatic conversion of human blood group A kidneys to universal blood group O

Serena MacMillan [1] ✉, Sarah A. Hosgood[1], Léonie Walker-Panse[1], Peter Rahfeld[2,3], Spence S. Macdonald [2,3], Jayachandran N. Kizhakkedathu [4,5], Stephen G. Withers [3] & Michael L. Nicholson [1] ✉

ABO blood group compatibility restrictions present the first barrier to donor-recipient matching in kidney transplantation. Here, we present the use of two enzymes, *Fp*GalNAc deacetylase and *Fp*Galactosaminidase, from the bacterium *Flavonifractor plautii* to enzymatically convert blood group A antigens from the renal vasculature of human kidneys to 'universal' O-type. Using normothermic machine perfusion (NMP) and hypothermic machine perfusion (HMP) strategies, we demonstrate blood group A antigen loss of approximately 80% in as little as 2 h NMP and HMP. Furthermore, we show that treated kidneys do not bind circulating anti-A antibodies in an ex vivo model of ABO-incompatible transplantation and do not activate the classical complement pathway. This strategy presents a solution to the donor organ shortage crisis with the potential for direct clinical translation to reduce waiting times for patients with end stage renal disease.

ABO blood group antigens are the most immunogenic of all the known blood grouping systems. Thus, specific compatibility restrictions between donors and recipients are put in place in blood transfusion and solid organ transplantation to prevent dangerous immunological reactions. In kidney transplantation, donor blood group antigens expressed on the surface of cells of the graft must be compatible with the native antibodies in the recipient to prevent hyperacute antibody-mediated damage to the organ, with the worst outcomes resulting in graft necrosis and rejection. These restrictions are summarised in Fig. 1a.

However, individuals of blood group O or B waiting for a kidney are disadvantaged in terms of access to compatible grafts with the average waiting time in the UK being two to three times longer for these patients compared to those of blood group A and AB[1]. This is due to the lower percentage of ABO-compatible donors for individuals of more restrictive blood types. Strategies to mitigate this wide gap in access to compatible transplants have been largely limited, with ABO-incompatible (ABOi) transplants providing one of the few clinically viable options. However, the requirement for pre-transplant

desensitisation and plasmapheresis to lower the anti-blood group antibody titre to acceptable levels requires a pre-planned transplant with a live donor[2].

Due to the increasing use of deceased donor kidneys, growing interest has focused on strategies of overcoming the ABO barrier in live and deceased donor grafts. One promising emerging strategy in solid organ transplantation involves enzymatic blood group conversion of an immunogenic graft to universal blood group O. The principle of the strategy involves using machine perfusion to perfuse an organ outside the body with bacteria-derived glycoside hydrolase enzymes that modulate the immunogenic blood group antigens expressed on the surface of the vascular endothelium. This converts the organ to a non-immunogenic state, permitting transplantation without risk of hyperacute antibody-mediated rejection. Such strategies have been investigated for blood group A to O conversion in human lungs, and B to O conversion in human kidneys[3,4]. While blood group A is the second most common blood group in the UK after blood group O, group A to O conversion in human kidneys has not been investigated until now.

[1]Department of Surgery, University of Cambridge, Cambridge, UK. [2]Avivo Biomedical Inc., Vancouver, BC, Canada. [3]Department of Chemistry, University of British Columbia, Vancouver, BC, Canada. [4]Department of Pathology and Laboratory Medicine, Centre for Blood Research, Life Sciences Institute, University of British Columbia, Vancouver, BC, Canada. [5]The School of Biomedical Engineering, University of British Columbia, Vancouver, BC, Canada. ✉e-mail: sgmjm2@cam.ac.uk; mln31@cam.ac.uk

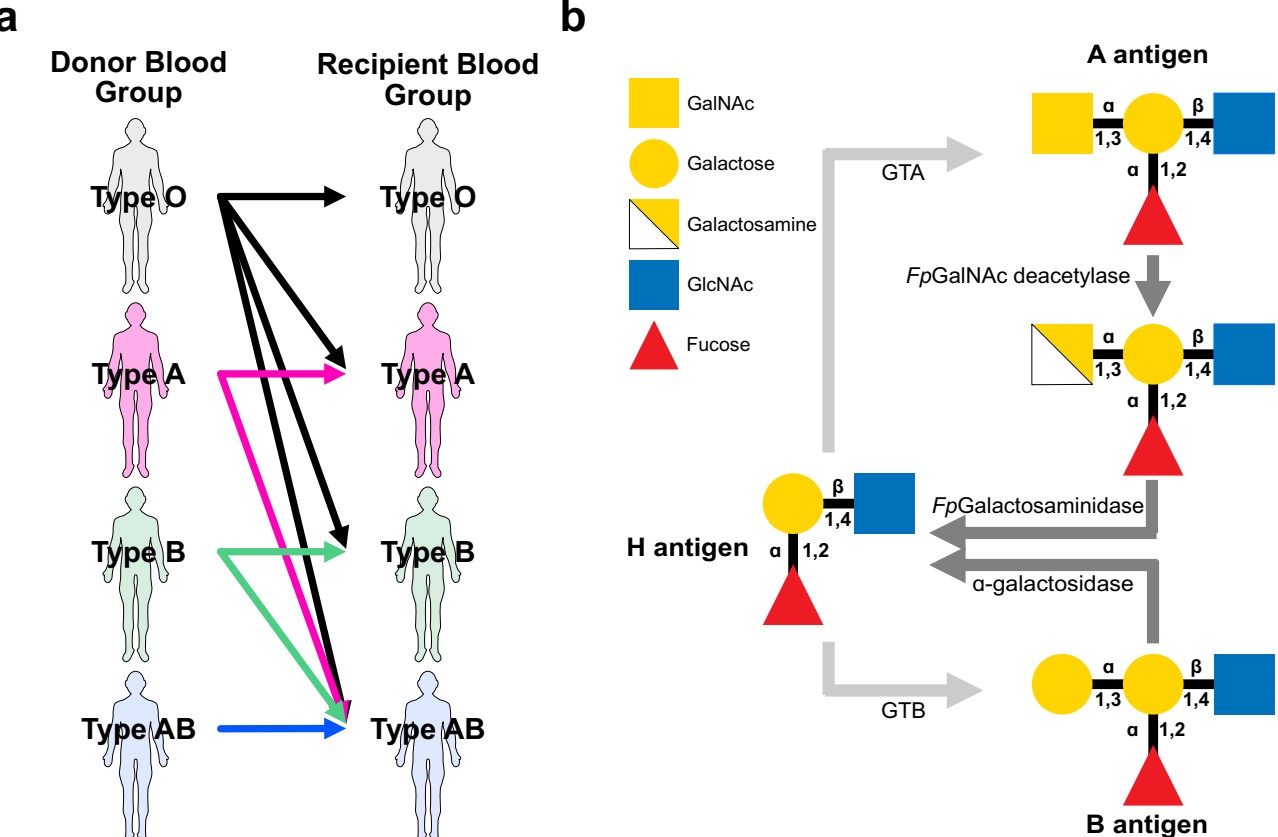

**Fig. 1 | ABO blood group compatibility and structure overview. a** ABO compatibility restrictions in transfusion and transplantation between donors and recipients of all four ABO blood groups (O, A, B, and AB). Arrowheads point from donors to ABO-compatible recipients. **b** Overview of blood group A or B antigen biogenesis from the core H antigen glycan by the action of the enzyme GTA or GTB, respectively. Enzymatic conversion from A or B antigens back to the H antigen by the action of *Fp*GalNAc deacetylase and *Fp*Galactosaminidase, or α-galactosidase, respectively, are shown by the darker grey arrows. Symbols (above) and numbers (below) within glycan structures indicate the type of linkage between monosaccharides. Type 2 antigen structures are shown. All structures follow the standardised Symbol Nomenclature for Glycans[48,49]. GalNAc N-acetylgalactosamine, GlcNAc N-acetylglucosamine, GTA blood group A N-acetylgalactosaminyltransferase, GTB blood group B galactosyltransferase.

All studies investigating enzymatic blood group conversion of human organs to date have used normothermic machine perfusion to deliver the enzymes to the vasculature of the organs. Such machine perfusion strategies for therapeutic intervention are attractive due to the direct targeting of the organ of interest without potentially harmful systemic effects in the patient[5–7]. Indeed, most research interest has focused on interventions during normothermic perfusion because hypothermic perfusion suppresses cellular activity, making it challenging to achieve effective cellular responses or pharmacodynamics. Despite this, hypothermic perfusion is the only perfusion strategy fully established in clinical practice and thus represents a promising avenue to explore blood group conversion of organs for transplantation.

In this report, we outline the use of two enzymes derived from the bacterium *Flavonifractor plautii* to convert human blood group A kidneys to universal blood group O using normothermic and hypothermic machine perfusion technologies. *Fp*GalNAc deacetylase and *Fp*Galactosaminidase work sequentially to convert the terminal N-acetyl-galactosamine of blood group A antigens to a galactosamine which is then removed to form the core H antigen found in all blood groups, including blood group O[8] (Fig. 1b). We then describe a model of ABOi reperfusion of pairs of type A kidneys, where enzyme-treated kidneys are modelled against their untreated biological pairs. This work provides a clinically feasible strategy for overcoming the ABO compatibility barrier in kidney transplantation.

## Results

### Donor demographics

In total, 18 donor kidneys from nine donors were declined for human transplantation and consented for research and recruited to this study between July 2022 and May 2023. All kidneys were blood group A, and the donor details are summarised in Table 1. In total, six kidneys (three biological pairs) were assigned to each of three research cohorts, namely, normothermic machine perfusion (NMP cohort), hypothermic machine perfusion (HMP cohort), or hypothermic machine perfusion followed by ABO-incompatible reperfusion (HMP-ABOi cohort).

### Enzymatic blood group A antigen removal during normothermic machine perfusion

Previous studies have shown *Fp*GalNAc deacetylase and *Fp*Galactosaminidase, herein referred to as *Fp*GalNAc DeAc and *Fp*GalNase, can remove blood group A antigens from human lungs during normothermic perfusion[4]. We sought to investigate A antigen removal in human kidneys by perfusing three biological pairs of blood group A kidneys for 6 h using an acellular perfusate at approximately 37 °C. These kidneys were designated the 'NMP cohort' and the demographics of the kidney donors are shown in Table 1, with all three pairs of kidneys retrieved from DCD donors. Within each pair, one kidney was treated with 1 mg/L of each of *Fp*GalNAc DeAc and *Fp*GalNase administered via the arterial cannula, and the contralateral kidney acted as a control with no enzyme addition. Treated kidneys were termed 'ABOe' kidneys.

**Table 1 | Donor demographics for the pairs of kidneys in the NMP, HMP, and HMP-ABOi cohorts**

| | NMP cohort (n = 3) | HMP cohort (n = 3) | HMP-ABOi cohort (n = 3) | All pairs of kidneys (n = 9) |
|---|---|---|---|---|
| Donor age (median; IQR) | 68, 67, 75 | 77, 37, 71 | 54, 54, 69 | 68 (54–73) |
| Male | 2 | 1 | 1 | 4 |
| Female | 1 | 2 | 2 | 5 |
| Blood group A+ | 1 | 1 | 2 | 4 |
| Blood group A- | 2 | 2 | 1 | 5 |
| DBD | 0 | 1 | 2 | 3 |
| DCD | 3 | 2 | 1 | 6 |
| WIT (min) | 7, 14, 9 | 6, 16 | 11 | 8 ± 6 |
| CIT (min) | 1023, 1090, 594 | 1314, 899, 1562 | 1558, 1292, 1088 | 1157 ± 312 |
| Terminal creatinine (µmol/L) | 77, 56, 48 | 64, 98, 99 | 31, 45, 145 | 74 ± 35 |
| Cause of death: | | | | |
| Intracranial haemorrhage | 2 | 3 | 2 | 7 |
| Hypoxic brain injury | 1 | 0 | 1 | 2 |
| Reason for decline for KTx: | | | | |
| Anatomy | 1 | 3 | 0 | 4 |
| PMH | 0 | 0 | 1 | 1 |
| Suspected malignancy in donor | 2 | 0 | 1 | 3 |
| Ischaemic bowel | 0 | 0 | 1 | 1 |

Donor details for all nine kidney pairs are summarised in the right-most column. *N* value refers to number of pairs of kidneys. Each value in a list refers to a unique pair of kidneys from a single donor. Summary values are mean ± standard deviation unless otherwise described in the table. *NMP* normothermic machine perfusion, *HMP* hypothermic machine perfusion, *ABOi* ABO-incompatible, *IQR* inter-quartile range, *DBD* donors after brain death, *DCD* donors after circulatory death, *WIT* warm ischaemic time, *CIT* cold ischaemic time, *KTx* kidney transplant, *PMH* past medical history.

Biopsies taken during perfusion were analysed by immunofluorescence microscopy for the presence of the A and H antigens (Fig. 2a). Quantification of A antigen removal and H antigen emergence during perfusion are shown in Fig. 2b for six fields of view from the three kidneys per group ($n = 18$ images per group total). A summary of normalised A and H antigen expression quantification is shown in Tables 2 and 3, respectively.

Pre-treatment sections show strong peritubular capillary staining for the A antigen, with some mild H antigen expression on the vascular surface (Fig. 2a). After 1 h, blood group A antigen expression in ABOe kidneys dropped to 56% of the pre-perfusion expression level, which was significantly lower than the control group ($p = 0.0056$). Blood group A antigen expression reached a low of 19% after 2 h in ABOe kidneys, with no further decrease observed up to 6 h. This was significantly lower than control kidneys which showed no decrease in antigen expression during perfusion.

In contrast, H antigen expression increased by 4.85-fold after 1 h *Fp*GalNAc DeAc and *Fp*GalNase treatment in the ABOe group, in line with the conversion of the A antigen structure to that of the H antigen (Fig. 1b), with no further change at 6 h. H antigen expression in the control group showed no increase over the course of perfusion.

**Effect of *Fp*GalNAc DeAc and *Fp*GalNase treatment on renal function during NMP**

Functional perfusion parameters were assessed continuously during perfusion for signs of treatment-related kidney injury. Renal blood flow (RBF) and mean arterial pressure (MAP) were stable throughout

the perfusion, with no difference between control and treated kidneys across the perfusion period of 6 h (Fig. 2c, d). After correction for individual kidney weight, mean RBF at 6 h was $230 \pm 76$ ml·min$^{-1}$ 100 g$^{-1}$ (mean ± SD) for control kidneys compared to $249 \pm 33$ ml·min$^{-1}$ 100 g$^{-1}$ for those treated with *Fp*GalNAc DeAc and *Fp*GalNase ($p = 0.7500$). MAP was also comparable between the groups (missing value prevents statistical testing). Urine output was also measured hourly during perfusion, with no significant difference between mean total urine output across the 6 h of perfusion (control: $46 \pm 10$ ml; treated: $37 \pm 28$ ml; $p = 0.7500$; Fig. 2e).

To evaluate potential histological signs of damage induced by the *Fp*GalNAc DeAc and *Fp*GalNase treatment, haematoxylin and eosin (H&E) stained biopsies were assessed pre- and post-treatment and showed no difference in injury grading between control and ABOe kidneys. Mild acute tubular injury was noted in pre-perfusion as well as post-perfusion samples in all cases, with vacuolation, fibrosis, cell shedding, and tubular flattening highlighted as signs of ischaemic injury attributed to the period of cold ischaemia after organ retrieval (Supplementary Fig. 1). No signs of acute vascular damage were indicated in treated or control samples.

Furthermore, the kidney damage marker NGAL was assessed after 6 h perfusion, with no significant difference found in perfusate (control group = $158 \pm 70$ ng/ml; treated = $205 \pm 114$ ng/ml; $p = 0.5000$) or urinary NGAL concentration between control and treated groups (control group = $133 \pm 85$ ng/ml; treated = $181 \pm 123$ ng/ml; $p = 0.2500$; Fig. 2f).

**Enzymatic blood group A antigen removal during hypothermic machine perfusion**

After successfully achieving antigen removal during NMP, we investigated the possibility of using the enzymes during hypothermic machine perfusion (HMP) which is the simplest perfusion strategy to integrate into clinical practice. We first aimed to examine the efficacy of antigen removal during HMP over a 24 h perfusion period using a LifePort Kidney Transporter at approximately 4 °C. Six kidney pairs were perfused, where one kidney per pair was treated as before with *Fp*GalNAc DeAc and *Fp*GalNase, and the other kidney acted as a control. Three kidney pairs were perfused for the full period of 24 h (HMP cohort), and a further three pairs for 6 h (HMP-ABOi cohort). Biopsies taken at various timepoints (pre, 1 h, 6 h, 12 h, 24 h for HMP cohort and pre, 1–6 h for HMP-ABOi cohort) were analysed by immunofluorescence microscopy (Fig. 3a; Supplementary Fig. 2).

No significant change in blood group A antigen expression was found after 1 h of *Fp*GalNAc DeAc and *Fp*GalNase perfusion compared to untreated controls ($p = 0.2856$; Fig. 3b; Table 2). A significant decrease in expression compared to the control group was noted after 2 h ($p = 0.0015$), with a maximal antigen decrease to approximately 20% expression level, which was maintained up to 24 h (Fig. 3b).

H antigen expression increased by 3.36-fold after 1 h in the ABOe group compared to the slight increase in the control group (1.87-fold; $p = 0.0204$; Table 3). H antigen expression reached peak expression at 4 h, decreasing up to 24 h although was significantly higher than the control group at all timepoints. The increase in H antigen expression in ABOe kidneys was markedly higher during HMP than in the NMP group, reaching an approximate maximum of 35.0-fold after 4 h HMP, compared to the maximal 4.85-fold increase in the NMP group.

**Effect of *Fp*GalNAc DeAc and *Fp*GalNase treatment on renal function during HMP**

Histological assessment of H&E-stained serial biopsies again showed signs of acute tubular injury in pre- and post-perfusion samples, but no signs of additional vascular injury or progressing ischaemic changes up to 24 h perfusion (Supplementary Fig. 3). Assessment of NGAL concentration in the circulating perfusate after 6 h of machine perfusion of control and treated kidneys showed no significant difference (control = $30 \pm 28$ ng/ml; treated = $23 \pm 13$ ng/ml; $p = 0.5625$;

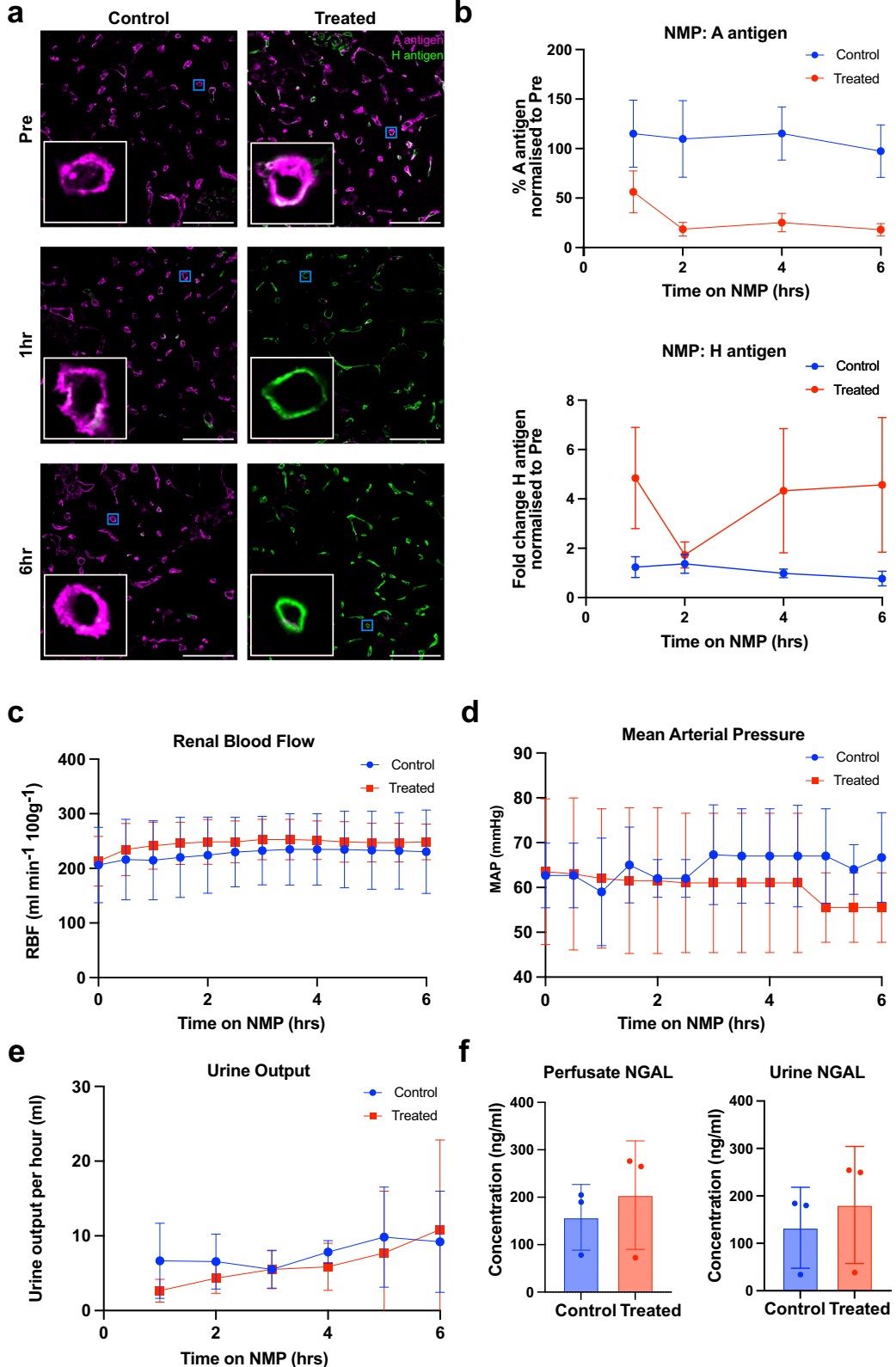

**Fig. 2 | Blood group A antigen removal during normothermic machine perfusion (NMP). a** Immunofluorescence images of blood group A antigen removal during NMP in control (left column) or treated (right column) type A human kidney cortical biopsies. At each timepoint (pre-treatment, 1 h after enzyme addition, or 6 h after enzyme addition), a composite image of blood group A antigens (magenta) and H antigens (green) is shown. An 8× zoom inlay of a representative peritubular capillary (blue box) is shown in the bottom left of each panel (white box). Scale bar represents 100 μm. **b** Quantification of A and H antigen expression at 1 h, 2 h, 4 h, and 6 h after enzyme addition in control (blue) and treated (red) kidneys. Antigen expression is normalised to expression level in the pre-perfusion biopsies for $n = 3$ kidney pairs. Error bars show mean ± 95% confidence intervals. Renal blood flow (RBF; panel **c**), mean arterial pressure (MAP; panel **d**) and urine output (**e**) during perfusion are shown for control and treated kidneys. Error bars show mean ± standard deviation. **f** The concentration of perfusate and urine NGAL after 6 h perfusion in control and treated kidneys. Error bars show mean ± standard deviation. NMP normothermic machine perfusion, NGAL neutrophil gelatinase-associated lipocalin.

Supplementary Fig. 4). These values were notably lower than the NMP cohort due to the suppressed metabolism in hypothermic vs normothermic conditions.

Functional parameters were continuously assessed, with no significant difference observed between control and treated kidneys in terms of RBF at 6 h ($p = 0.9375$), or 24 h ($p = 0.5000$; Fig. 3c). Similarly, no significant difference was observed for MAP at 6 h ($p = 0.2500$) or 24 h ($p > 0.9999$; Fig. 3d). During HMP, the cold perfusion solution is continuously recirculated from the central chamber containing the submerged kidney and so urine output is not monitored.

**Table 2 | Quantification of A antigen expression in control vs treated kidneys during NMP or HMP**

| Time after enzyme addition (h) | Percentage of A antigen remaining during NMP | | | Percentage of A antigen remaining during HMP | | |
|---|---|---|---|---|---|---|
| | Control (%) | Treated (%) | p | Control (%) | Treated (%) | p |
| 1 | 115 [81,148] | 56 [35,78] | **0.0056** | 77 [57,98] | 70 [46,95] | 0.2856 |
| 2 | 110 [71,148] | 19 [12,25] | **<0.0001** | 54 [43,64] | 23 [13,32] | **0.0015** |
| 3 | / | / | / | 62 [34,91] | 24 [11,37] | **0.0047** |
| 4 | 115 [88,142] | 25 [16,35] | **<0.0001** | 47 [31,62] | 20 [10,29] | **0.0047** |
| 5 | / | / | / | 38 [23,52] | 13 [7,21] | **0.0042** |
| 6 | 97 [71,124] | 18 [12,24] | **<0.0001** | 85 [67,103] | 21 [12,29] | **<0.0001** |
| 12 | / | / | / | 119 [69,168] | 29 [18,39] | **0.0005** |
| 24 | / | / | / | 96 [72,120] | 28 [18,38] | **0.0042** |

Values represent the mean percentage of A antigen expression in images from the specified timepoint relative to the pre-treatment biopsy in each experiment. 95% CI are shown underneath in square brackets. At each timepoint, data are pooled from a minimum of three independent kidney biopsies providing at least 18 fields of view. To compare expression in control and treated kidneys, a two-tailed Wilcoxon matched-pairs signed rank test was completed on each row, with Holm-Sidak's multiple comparisons correction. P-values are shown to 4 decimal places. Significant results ($p < 0.0500$) are highlighted in bold.
NMP normothermic machine perfusion, HMP hypothermic machine perfusion.

### Comparison of blood group A antigen removal during NMP vs HMP

A direct comparison of the kinetics of A antigen removal in kidneys treated during NMP vs HMP is summarised in the heat map in Fig. 4, where both strategies reached comparable levels of antigen removal after approximately 2 h. As enzyme treatment during HMP is the most clinically translatable strategy at present, we proceeded with further experiments using kidneys perfused during HMP for 6 h to ensure full conversion.

### ABOe kidneys do not bind circulating antibodies in an ABOi reperfusion model

After demonstrating that *Fp*GalNAc DeAc and *Fp*GalNase can remove blood group antigens during NMP and HMP, we sought to assess whether ABOe kidneys could withstand the immunological challenge of ABO-incompatible conditions in a potential recipient. We established an ex vivo model of an A to O transplant in the three pairs of human kidneys that underwent only 6 h HMP (HMP-ABOi cohort). Each kidney underwent 6 h HMP with or without *Fp*GalNAc DeAc and *Fp*GalNase treatment, as outlined previously, and were subsequently reperfused with perfusate containing anti-A and -B antibodies at normothermic temperature (37 °C). Briefly, each pair of kidneys was removed from the LifePort after the HMP phase, placed on ice, and flushed with Ringer's solution before undergoing a 4 h reperfusion phase with a type O red blood cell-based solution supplemented with 10% human AB serum. Human type AB serum contains no anti-A or anti-B antibodies, so to induce ABOi conditions, purified and concentrated mouse monoclonal anti-A and -B IgM antibody was manually added after 20 min of perfusion to a set final titre of 1:128 of anti-A and 1:128 anti-B. Human serum was used instead of fresh-frozen plasma (FFP) to prevent the presence of anticoagulants used in plasma collection from inhibiting complement activity, although this meant our circuit contained no coagulation factors.

Haemodynamic parameters during the ABOi reperfusion phase were similar in control and treated kidneys (Fig. 5). After 4 h perfusion, no significant difference was observed between incompatible control kidneys (ABOi) and enzyme-treated (ABOe) in terms of the RBF ($p = 0.2500$; Fig. 5a) or MAP ($p = 0.7500$; Fig. 5b) which is to be expected as our model did not include coagulation factors. Total urine output was numerically higher in the ABOe group, but this did not reach significance (ABOi = $42 \pm 15$ ml; ABOe = $74 \pm 36$ ml; $p = 0.5000$;

**Table 3 | Quantification of H antigen expression in control vs treated kidneys during NMP or HMP**

| Time after enzyme addition (h) | Fold-change of H antigen expression during NMP | | | Fold-change of H antigen expression during HMP | | |
|---|---|---|---|---|---|---|
| | Control | Treated | p | Control | Treated | p |
| 1 | 1.24 [0.82,1.66] | 4.85 [2.80,6.90] | **0.0354** | 1.87 [1.33,2.42] | 3.36 [2.09,4.63] | **0.0204** |
| 2 | 1.37 [0.99,1.75] | 1.73 [1.21,2.26] | 0.6112 | 0.45 [0.23,0.65] | 29.80 [14.30,45.30] | **<0.0001** |
| 3 | / | / | / | 0.85 [0.44,1.26] | 29.50 [15.90,43.10] | **<0.0001** |
| 4 | 0.98 [0.81,1.16] | 4.33 [1.82,6.85] | **0.0410** | 0.83 [0.46,1.20] | 34.90 [16.10,53.70] | **<0.0001** |
| 5 | / | / | / | 0.81 [0.41,1.22] | 28.00 [16.70,39.40] | **<0.0001** |
| 6 | 0.77 [0.47,1.07] | 4.57 [1.84,7.29] | 0.1048 | 1.18 [0.81,1.55] | 18.10 [12.20,24.00] | **<0.0001** |
| 12 | / | / | / | 1.42 [0.74,2.11] | 12.30 [6.51,18.10] | **<0.0001** |
| 24 | / | / | / | 0.61 [0.16,1.06] | 19.08 [11.70,26.50] | **<0.0001** |

Values represent the mean fold-change of H antigen expression in images from the specified timepoint relative to the pre-treatment biopsy in each experiment. 95% CI are shown in square brackets. At each timepoint, data are pooled from a minimum of three independent kidney biopsies providing at least 18 fields of view. To compare expression in control and treated kidneys, a two-tailed Wilcoxon matched-pairs signed rank test was completed on each row, with Holm-Sidak's multiple comparisons correction. P-values are shown to 4 decimal places. Significant results ($p < 0.0500$) are highlighted in bold.
NMP normothermic machine perfusion, HMP hypothermic machine perfusion.

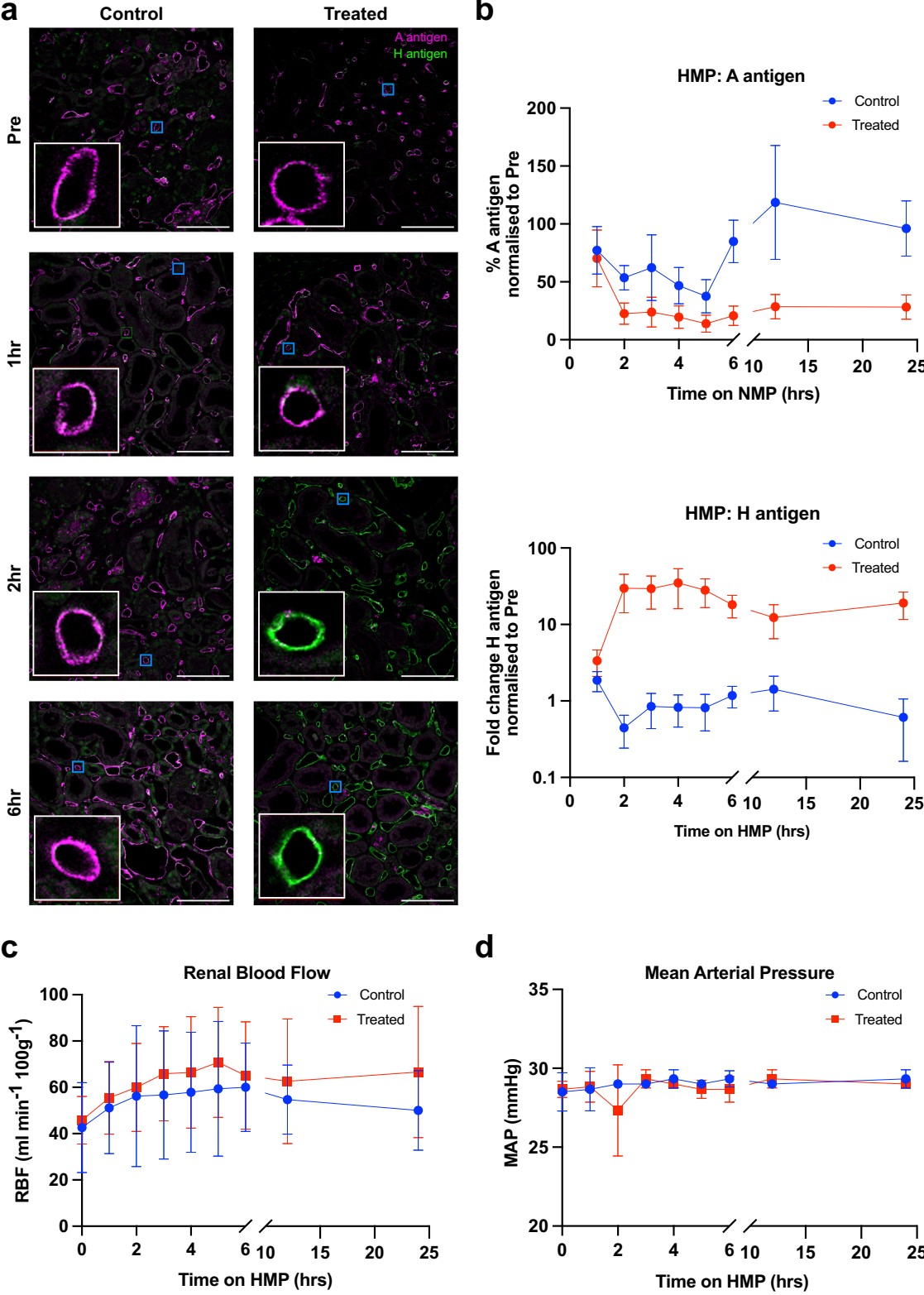

**Fig. 3 | Blood group A antigen removal during hypothermic machine perfusion (HMP). a** Immunofluorescence images of blood group A antigen removal during HMP in control (left column) or treated (right column) type A human kidney cortical biopsies. At each timepoint (pre-treatment, 1 h, 2 h or 6 h after enzyme addition) a composite image of blood group A antigens (magenta) and H antigens (green) is shown. An 8× zoom inlay of a representative peritubular capillary (blue box) is shown in the bottom left of each panel (white box). Images are representative of a minimum of $n = 3$ kidney pairs. Scale bar represents 100 μm. **b** Quantification of A and H antigen expression at 1–6 h, 12 h and 24 h after enzyme addition in control (blue) and treated (red) kidneys. Antigen expression is normalised to expression levels in the pre-biopsies. Error bars show mean ± 95% confidence intervals. Renal blood flow (RBF; panel **c**) and mean arterial pressure (MAP; panel **d**) during perfusion are shown for control and treated kidneys. Error bars show mean ± standard deviation. HMP hypothermic machine perfusion.

Fig. 5c). Estimated glomerular filtration rate (eGFR) was also comparable between ABOi and ABOe kidneys during reperfusion (Fig. 5d). The kidney injury marker NGAL was measured in the perfusate and urine after 4 h reperfusion with no significant difference observed between groups (perfusate: $p = 0.5000$; urine: $p = 0.2500$; Supplementary Fig. 5a). H&E injury grading showed no evidence of increased vascular injury or ischaemic damage in the ABOi group (Supplementary Fig. 5b).

To assess whether the circulating antibody in the reperfusion phase could bind the vascular surface of enzyme-treated ABOe

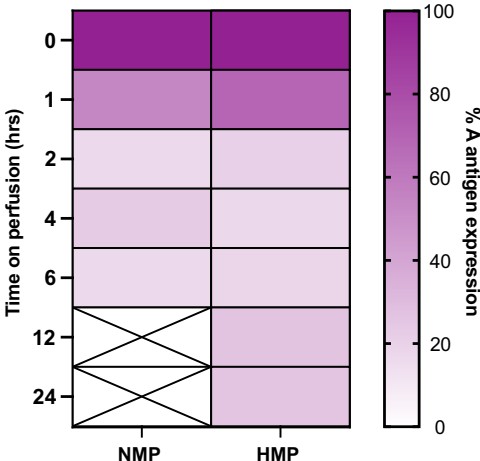

**Fig. 4 | Heat map summary comparing blood group A antigen loss in NMP vs HMP for *Fp*GalNAc DeAc and *Fp*GalNase-treated kidneys.** Y-axis indicates the hours on perfusion post-enzyme addition. Colour gradient represents percentage blood group A antigen expression from 100% (purple) to 0% (white) normalised to pre-perfusion levels (0 h). Data in each column are the mean expression level from $n = 3$ pairs except 1 h and 6 h timepoints in the HMP column which are derived from $n = 6$ kidneys. NMP normothermic machine perfusion, HMP hypothermic machine perfusion.

kidneys, biopsies were stained with fluorescently tagged goat anti-mouse-IgM (Fig. 6a). In ABOi kidneys, mouse IgM deposits were found on the surface of the peritubular capillaries in post-reperfusion biopsies, indicating binding in situ during active perfusion. Endothelial IgM deposition was significantly higher in all ABOi kidneys compared to their ABOe pairs where minimal binding was observed ($p < 0.0001$; Fig. 6b). This indicates that ABOe kidneys do not bind circulating anti-A antibodies in our model of ABO-incompatible reperfusion.

To further analyse antibody-antigen interaction in ABOi and ABOe kidneys, the relative amount of anti-A in the perfusate over time was assessed with a red blood cell flow cytometry assay. Briefly, samples of perfusate containing antibody were incubated with fixed human type A red blood cells (RBCs) which were then washed and incubated with a fluorescent anti-mouse-IgM antibody (AF647). The median fluorescence intensity (MFI) of serial samples was normalised to the MFI of samples taken immediately after antibody addition. Anti-mouse-IgM MFI decreased steadily during perfusion in the ABOi control group but remained constant in the ABOe group (Fig. 6c, d). Overall circulating antibody levels reached a low of $28.4\% \pm 7.4\%$ after 4 h perfusion in ABOi kidneys compared to $101.3\% \pm 21.6\%$ in ABOe kidneys. The findings are consistent with circulating antibody levels decreasing as anti-A antibodies bound free antigens on the vascular endothelium. ABOe kidneys retained sufficiently few blood group A surface antigens to have any effect on circulating anti-A antibody levels after 4 h, showing a promising feasible strategy of immune evasion of ABOi grafts in potential transplant recipients.

## Activation of the classical complement pathway in ABOi but not ABOe kidneys

As a feature of acute antibody-mediated rejection is activation of the classical complement pathway, activation components of the complement cascade were assessed in the perfusate and tissue of control (ABOi) and ABOe kidneys during reperfusion. The classical complement pathway is initially activated by IgM or IgG1 or IgG3 subclasses of IgG immunoglobulins binding their antigen on the cell surface and engaging the C1q-r$_2$s$_2$ complex[9]. A detailed summary of complement

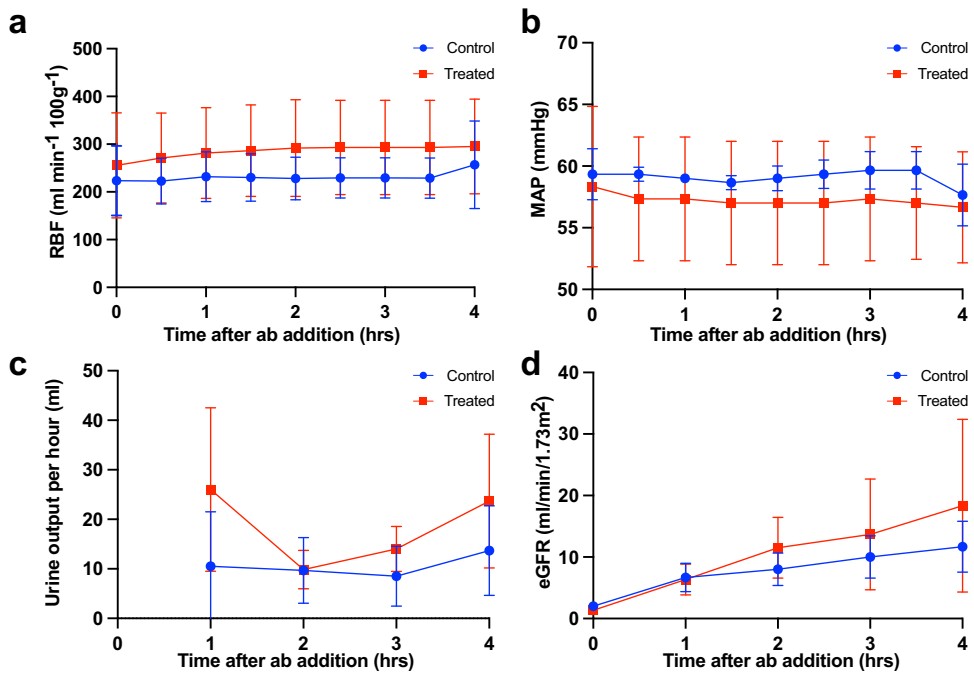

**Fig. 5 | Haemodynamic parameters during ABO-incompatible reperfusion.** Measurements of renal blood flow (RBF; **a**), mean arterial pressure (MAP; **b**), urine output (**c**), and eGFR (**d**) for control and treated kidneys during 4 h ABOi reperfusion for $n = 3$ kidney pairs. Error bars show mean ± standard deviation. eGFR estimated glomerular filtration rate, ABOi ABO-incompatible.

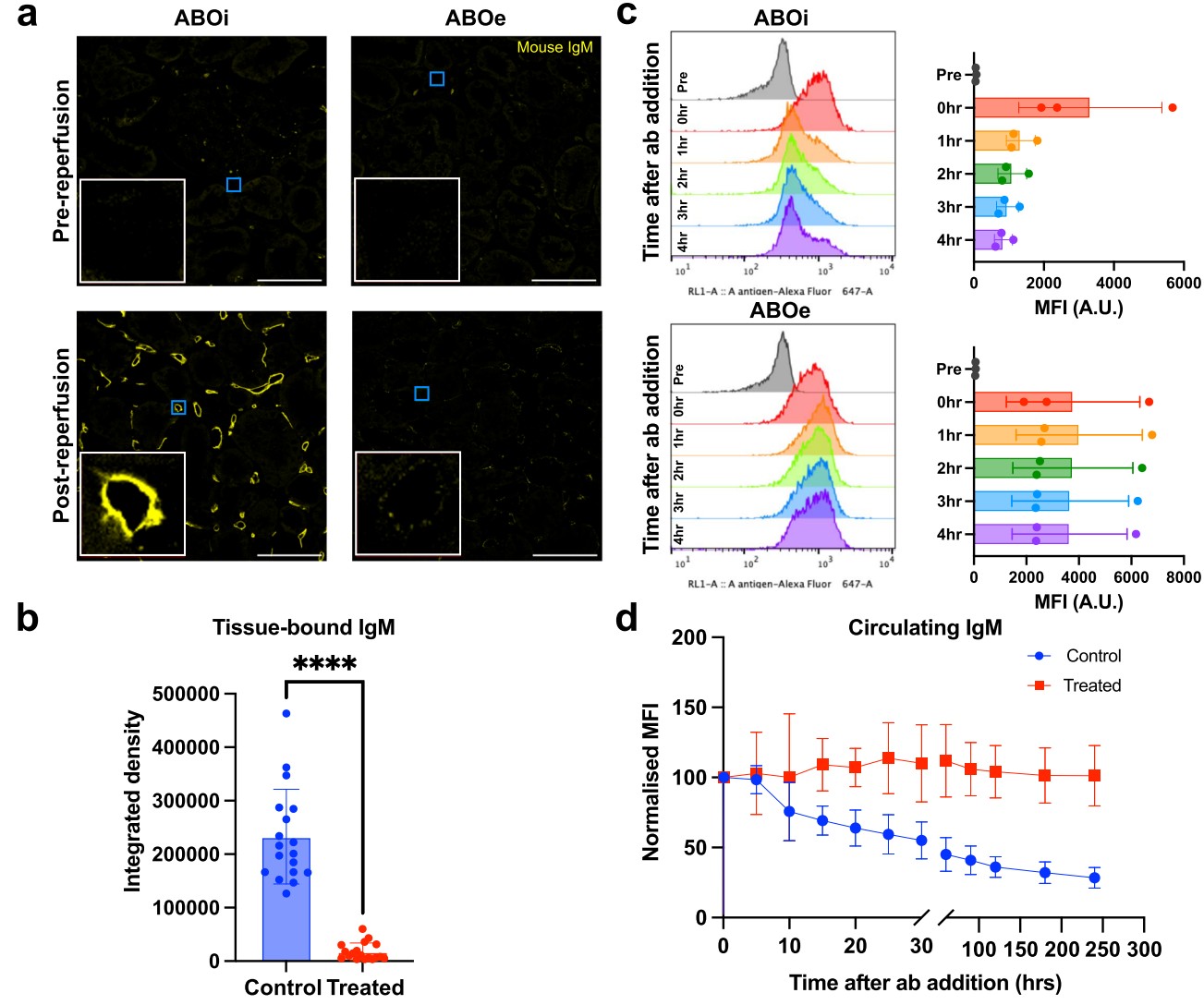

**Fig. 6 | Anti-A binding during ABO-incompatible reperfusion.**
**a** Immunofluorescence images of cortical kidney biopsies pre-reperfusion and 4 h post-reperfusion. Anti-IgM staining is shown in yellow with a representative capillary selected (blue box) for an 8× zoom inlay (white box). Capillaries in IgM negative sections were selected based on anti-H staining (separate channel, not shown). Scale bar shows 100 μm. **b** Quantification of IgM staining in control vs treated kidneys. Six fields of view were selected per kidney (*n* = 3) for 18 images total per group. Data were compared between control and treated groups with a two-tailed

Wilcoxon matched-pairs signed rank test. **c** Representative histogram of IgM-stained RBCs in perfusate samples taken before (pre), immediately after (0 h), and 1–4 h after antibody addition in one control (ABOi) and one treated (ABOe) kidney from a single biological pair. MFI of all kidneys at each timepoint is shown to the right per histogram. **d** MFI values at all time points normalised to 0 h MFI for *n* = 3 control and treated kidney pairs. In all cases, error bars show mean ± standard deviation. ****p < 0.0001. A.U. arbitrary units, RBCs red blood cells, MFI median fluorescence intensity.

---

activation is demonstrated in Fig. 7a. The presence of C1q, C4d, and C5b-9 (membrane attack complex) in the tissue was assessed by immunofluorescence microscopy while the concentration of the anaphylatoxin C5a and soluble C5b-9 (sC5b-9) were analysed in the perfusate and urine.

C1qA, a subunit of the C1q protein, was observed to bind the microvasculature in ABOi but not in ABOe tissue and directly colocalised with the presence of anti-A mouse IgM (Fig. 7b). Further activation components C4d and C5b-9 also showed discrete localisation to the microvasculature in ABOi but not ABOe tissue. There was no difference in the concentration of released soluble activation factors (C5a and sC5b-9) in the perfusate or urine between the two conditions at any timepoint, however (Supplementary Fig. 6). This may be due to the subtle changes in soluble component concentrations being masked by acute, non-sustained complement activation responses due to ischaemia reperfusion injury[10]. These results show firstly that the binding of mouse anti-A IgM in this model of ABOi reperfusion is

capable of engaging C1q binding to initiate activation of the classical complement cascade. Secondly, we can conclude that ABOe kidneys which do not bind circulating anti-A antibodies can avoid complement activation up to 4 h after reperfusion, demonstrating a key method of potentially evading hyperacute antibody-mediated rejection in a transplant setting.

## Discussion

This study has demonstrated the successful removal of blood group A antigens from human kidney vasculature in hypothermic and normothermic conditions. Moreover, we have established a model of ABO-incompatible transplantation using ex vivo machine perfusion and showed the initiation of the classical complement cascade via the engagement of C1q and the deposition of C4d and C5b-9. As an emerging strategy, human solid organ blood group conversion has only been reported twice before, once for blood group A removal using *Fp*GalNAc DeAc and *Fp*GalNase and NMP in lungs, and once for

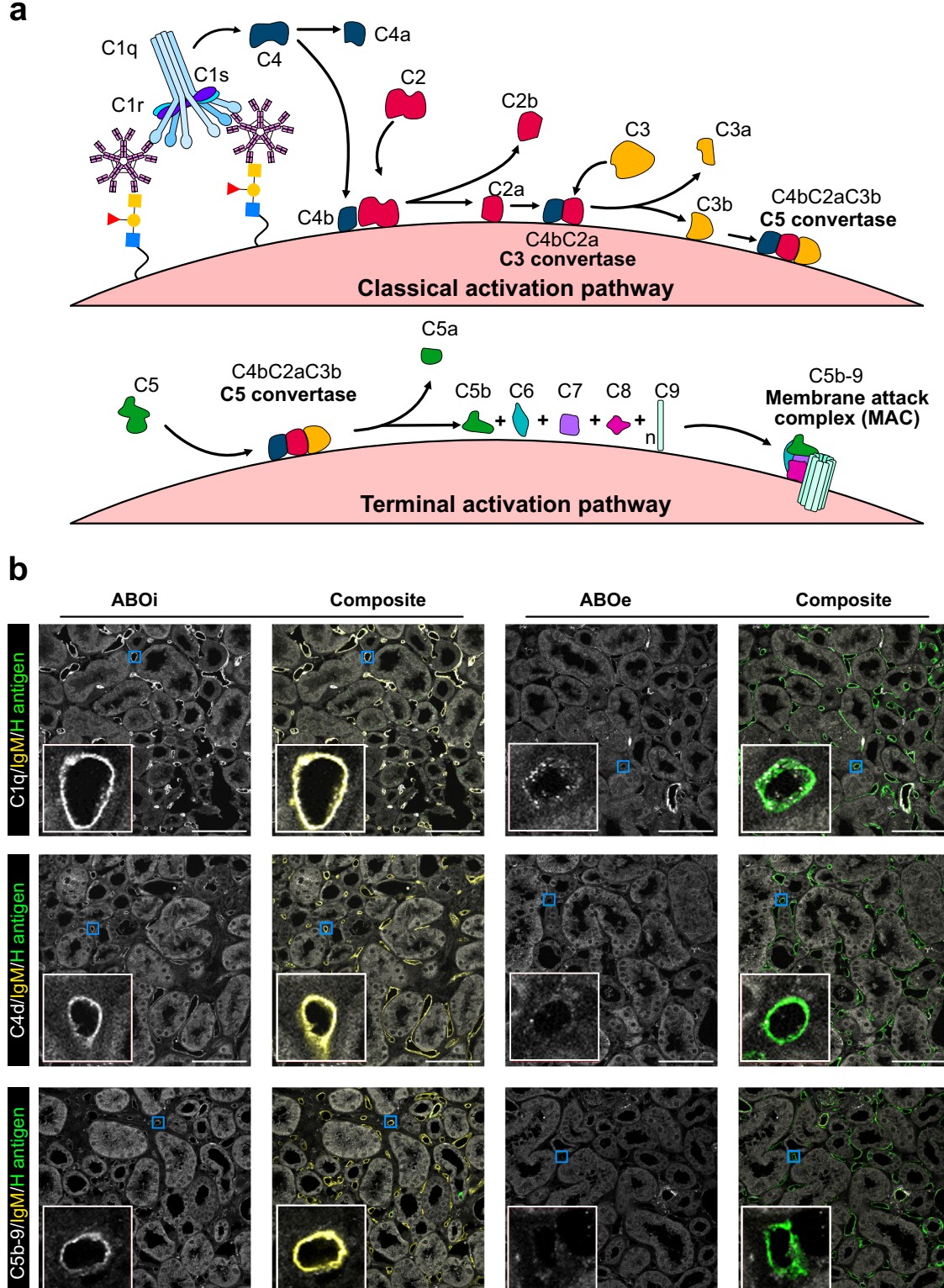

**Fig. 7 | Activation of the classical complement cascade during ABOi reperfusion. a** Schematic overview of the activation of the complement cascade, showing activation of the classical activation pathway (top panel), and terminal activation pathway (bottom panel). **b** Representative immunofluorescence images of kidney cortical biopsies after 4 h ABO-incompatible reperfusion in control (ABOi) and treated (ABOe) kidneys stained for C1q (top row; grey), C4d (middle row; grey), and C5b-9 (bottom row; grey) for $n = 3$ biological replicates. An 8× zoom inlay of a representative peritubular capillary (blue box) is shown in the bottom left of each panel (white box). Composite images also show IgM (yellow) and H antigen (green) staining. Scale bar represents 100 µm.

blood group B removal using an α-galactosidase (GH110B) during NMP in kidneys[3,4]. The study presented here represents the first report of blood group A removal in human kidneys, and the first report of enzymatic conversion using hypothermic machine perfusion. The ABOe approach could increase utility of donor organs already retrieved for transplantation to improve overall waiting times for kidney transplantation.

Here we show that both hypothermic and normothermic techniques converted approximately 80% of endothelial A antigens to the universal H antigen structure in human kidneys in as little as 2 h, demonstrating the viability of the technique for direct clinical translation. This also emphasises the flexibility of enzymatic blood group conversion which is achievable across different types of machine perfusion strategies and devices, and naturally lends itself to implementation for the conversion of other solid organs such as the liver or heart. It is interesting to note, however, that the rate of antigen conversion observed in these ABOe kidneys is lower than that reported in ABOe lungs which achieved 97% A antigen loss on pulmonary endothelial cells[4]. This is likely due to differences in quantification technique, where our method of fluorescence quantification of fixed tissue is less specific than flow cytometry of isolated cells and can be influenced by background A antigen expression found in distal convoluted tubules[11,12]. This would suggest our results are likely to be an underestimation of the success of vascular A antigen removal in kidneys.

It is of particular interest that we have shown the *Fp*GalNAc DeAc and *Fp*GalNase can function effectively during HMP. While NMP has been shown to be safe and feasible for human kidney transplantation in a recent randomised control trial[13], clinical implementation of organ perfusion as standard of care has only been achieved for hypothermic machine perfusion. Indeed, HMP has been shown to reduce delayed graft function and primary non-function compared to static cold storage in a recent meta-analysis[14]. Despite the clinical benefits demonstrated by HMP, little work has investigated the role of therapeutic interventions during hypothermic perfusion[15–19]. This is largely due to the colder temperature resulting in a low metabolic rate maintained in cells, yet HMP presents a promising avenue for intervention due to the wide clinical use and smaller, transportable perfusion machines which are commercially available. The ABOe strategy could be implemented into clinical practice during transport of organs from donor to recipient centre using a portable perfusion device such as the LifePort Kidney Transporter from Organ Recovery Systems used in this study.

We have also shown that *Fp*GalNAc DeAc and *Fp*GalNase treatment does not induce kidney injury, with no observed difference in haemodynamic parameters or NGAL concentration observed between control or treated cohorts during NMP or HMP. No histological signs of additional vascular damage were observed in either cohort in both the treated and control groups, indicating that the use of *Fp*GalNAc DeAc and *Fp*GalNase is safe in the acute setting using our human model. Indeed, the use of deceased donor human organs offers a rare advantage to the preclinical testing of novel therapies as there is no translational barrier associated with animal models.

In this study, all recruited kidneys were of blood group A which was determined by serological phenotyping of the donor. As ABO subtypes of deceased organ donors are not routinely screened in the UK, the blood group A subtype of the kidneys used in this study were not known prior to recruitment. This has important implications for understanding enzyme efficacy across different subtypes as A1 individuals express approximately five times more blood group A antigens on red blood cells than the rarer A2 subtype[20,21]. Limited quantitative data about subtype antigen expression in the renal endothelium is available, although lower expression has been indicated in A2 biopsies[11,22]. In the UK, A1 individuals represent approximately 80% of the blood group A population, thus most kidneys examined here are likely to be A1, although the variation of antigen loss between kidneys

during perfusion may be partially explained by these donor differences[23]. Further study of ABO genotypes and donor organ antigen expression may also reveal differences between homozygotes and heterozygotes which may influence future clinical decisions regarding organ allocation or ex vivo treatment.

The ABOe strategy presented here is one method of overcoming the ABO barrier in transplantation, although paired kidney exchange and widening allocation policy to include lower-risk A2 ABOi transplants have also been successful in lowering waiting times for more restrictive blood types[24–26]. Further development of these options, such as the inclusion of deceased donors to instigate a paired kidney exchange chain, may also improve access to living kidney donation[27,28], although such programmes are not available in some countries due to ethical concerns.

In our model of ABO-incompatibility, we demonstrated that ABOe kidneys do not activate the classical complement cascade, whereas untreated kidneys showed deposition of activated complement markers. This is the first evidence that the ABOe strategy in kidneys can evade humoral responses in a model of an ABO-incompatible recipient. Key among these markers is C4d. Peritubular C4d staining is considered by the Banff classification of allograft pathology as a feature of antibody-mediated rejection (ABMR) but its diagnostic value in the absence of other histological features has been called into question[29–32]. In 2013, the classification was revised to include C4d-negative ABMR, where microvascular injury was found in the presence of donor-specific antibodies but in the absence of C4d deposition[33]. The utility of C4d as a biomarker of ABMR in ABOi biopsies is also contested, with multiple studies showing little histological signs of injury associated with C4d deposits in post-ABOi kidney transplant biopsies[34–36]. Other studies have shown that the presence of C4d in association with microvascular injury is indeed suggestive of ABMR in ABOi recipients[37–39]. In the present study, the deposition of C4d in control ABOi kidneys demonstrates the capacity for complement activation in our model of ABOi conditions, and, most importantly, shows that enzymatic removal of immunogenic ABO antigens can prevent such activation. The predictive or functional consequences of this early complement response will need to be studied further to understand the immunological risk associated with an ABOe transplant.

An essential consideration to this work is the kinetics of antigen re-emergence after removal. The current strategy presented here provides an immediate method to remove the antigenicity of an incompatible donor graft for transplantation, with the main benefit found in re-allocation of the deceased donor pool in favour of more restrictive blood groups. This does not, however, preclude the possibility of antigen re-emergence and subsequent engagement by the recipient innate immune system in the post-transplant phase. While the kinetics of antigen re-establishment are unknown at present, it is expected to occur within an acute timeframe, likely on the scale of days or weeks. However, it is also theorised that re-emergence may not present a graft-threatening immunological event due to the potential effects of a phenomenon called accommodation. This is defined as the immunological tolerance to otherwise incompatible antigenic conditions in transplantation and has been widely characterised in ABOi recipients[40,41]. In these patients, grafts expressing incompatible antigens are tolerated despite the re-emergence of anti-A or -B circulating antibodies after they have been removed by pre-transplant plasmapheresis. Future preclinical work using extended perfusion strategies or animal models will be required to elucidate the functional consequences of antigen re-emergence in the recipient.

The study presented here has some key limitations, primarily the small study size. This is largely owing to the availability of appropriate blood group A donor kidney pairs consented for research. However, we have successfully established conditions for the effective removal

of the blood group A antigen in the human kidney in all nine treated grafts examined. Moreover, the study is strengthened by the exclusive use of biological pairs of human kidneys in each case, providing a perfect control for each experiment, matched exactly for donor age, sex, co-morbidities, and genetic variation.

A further limitation of the presented study is the lack of platelets or clotting factors within the circulating perfusate in the ABOi model. The perfusate was supplemented with human off-the-clot serum as a source of soluble complement factors and metabolites to enable activation of the complement pathway but is unable to activate the coagulation cascade, a key process in hyperacute ABMR. This is reflected in the absence of haemodynamic changes and histological signs of microthrombi in the ABOi kidneys during reperfusion, as noted in an ex vivo ABOi model established by Chandak et al.[42]. This choice was to prevent the introduction of anticoagulants such as citrate to the perfusate. During the collection of whole blood prior to plasma separation, anticoagulants chelate the divalent metals ions $Ca^{2+}$ and/or $Mg^{2+}$ to prevent clotting. However, this chelation also prevents $C1q$-$C1r_2$-$C1s_2$ and $C4b$-$C2$ complex formation, respectively[43,44]. The use of human serum over plasma is therefore required for a study of complement function[45]. This choice also allowed for reproducible blood group antibody titres (1:128) in the perfusate through the addition of monoclonal mouse IgM which is difficult to achieve with human fresh-frozen plasma due to the variability of donor antibody levels.

A final limitation is the short reperfusion time of four hours in the ABOi model which was chosen to represent the hyperacute post-implantation phase of an ABOi transplant. Despite this, we were still able to observe the critical tissue-antibody interaction and the downstream complement activation in the ABOi group, although functional readouts of kidney damage may be more pronounced at a later timepoint. Advances in prolonged kidney perfusion will enable further studies to investigate this further, up to 12 h or 24 h post-treatment.

In summary, this research has shown antigen removal in a clinically viable timeframe of two hours in both strategies of machine perfusion. Our work has the potential to cause dynamic change in solid organ allocation by removing the ABO barrier in transplantation.

## Methods

### Study population
Eighteen human donor kidneys from nine deceased donors of blood group A were recruited to this study after being declined for transplantation and consented for research. Written consent for the use of kidneys for research was given by the donor families and was obtained by Specialist Nurses in Organ Transplantation. Ethical approval was obtained from NRES: 15/NE/0408 and 22/WA/0167.

### Normothermic machine perfusion (NMP)
Three pairs of kidneys (NMP cohort) were perfused for 6 h with an oxygenated acellular perfusate at approximately 37 °C using a modified method from Nicholson and Hosgood[46]. A detailed protocol is provided in the Supplementary Information. Briefly, the kidneys were perfused at approximately 37 °C with 500 ml acellular perfusate (see Supplementary Information for the full composition) using an adapted paediatric cardiopulmonary bypass machine. After a stabilisation period of 20 min, the treated kidney of the pair was injected with 0.5 ml each of *Fp*GalNAc deacetylase and *Fp*Galactosaminidase, each at a concentration of 1 mg/ml, via the arterial cannula (time 0 h). Renal blood flow (RBF) and mean arterial pressure (MAP) were recorded every 30 min with urine output recorded hourly. Arterial perfusate and urine samples were collected hourly, flash-frozen in liquid nitrogen and stored long-term at −80 °C. Biopsies were taken at the specified time points and were formalin-fixed and paraffin-embedded (FFPE).

### Hypothermic machine perfusion (HMP)
Three pairs of kidneys were perfused at approximately 4 °C for 24 h using a LifePort Kidney Transporter (Organ Recovery Systems, Itasca, IL, USA) as per the manufacturer's recommendations. Briefly, each pair of kidneys was immersed in a LifePort cassette containing 1 L of Belzer MPS UW Machine Perfusion Solution and perfused for a 20 min stabilisation period before the treated kidney was injected with 1 ml of each of *Fp*GalNAc deacetylase and *Fp*Galactosaminidase, each at a concentration of 1 mg/ml, via the arterial cannula (time 0 h). Kidneys were perfused for 24 h and monitored for RBF and MAP and sampled at the specified timepoints. Biopsies and samples of perfusion solution were also taken at these time points and stored as described for the NMP cohort. Three further pairs of kidneys underwent 6 h HMP (±*Fp*GalNAc deacetylase and *Fp*Galactosaminidase) in preparation for ABO-incompatible reperfusion (see next section). The kidneys were then removed from the LifePort Kidney Transporter, placed on ice, and flushed with 500 ml Ringer's solution via the arterial cannula and the ureter was cannulated.

### ABOi reperfusion model
Briefly, the ABOi perfusate per kidney contained a similar composition to the NMP perfusate supplemented with one unit of type O RBCs and 50 ml off-the-clot type AB human serum (TCS Biosciences Ltd., Buckingham, UK). The full composition is described in the Supplementary Information. The kidneys were perfused for a stabilisation period of 20 min. At this time, a biopsy and arterial perfusate sample were taken as an antibody-free baseline of normothermic perfusion. Then, 31.25 ml of concentrated mouse anti-A and -B IgM combined (titre 1:2048 in each case; Lorne Laboratories Ltd, Reading, UK) was added to each kidney in each pair, providing a final titre of 1:128 in the 500 ml perfusate and was termed time 0 h. Arterial perfusate samples, RBF readings, and MAP readings were taken at the specified time points. Biopsies were taken at the specified time points, and hourly urine samples and urine output volume readings were also taken. Perfusate samples for downstream complement analysis were taken in EDTA blood tubes. All samples were flash-frozen in liquid nitrogen and stored long-term at −80 °C.

### Biochemical analysis
For the NMP and reperfusion cohorts, fresh arterial perfusate and urine samples taken hourly were sent to the NIHR Clinical Biochemistry facility at Addenbrooke's Hospital for measurements of serum creatinine, urine creatinine, eGFR, and electrolytes.

### Haematoxylin and eosin (H&E) staining and histological analysis
Clinical grade H&E staining was completed by the Human Research Tissue Bank at Addenbrooke's Hospital on 4 µm FFPE sections. All H&E-stained sections were imaged using an Olympus IX81 inverted microscope with a 10× objective (Olympus Corporation, Tokyo, Japan). A minimum of five fields of view per section were imaged. A consultant renal pathologist provided an injury report on representative sections from each timepoint of each experiment.

### Immunofluorescence staining and microscopy
FFPE biopsies were cut to 4 µm sections for immunostaining. All sections were deparaffinised with xylene and dehydrated with an ethanol gradient before heat-induced epitope retrieval using a Tris-EDTA buffer at pH 9.0. Sections were stained with primary antibodies for 1 h at room temperature in PBS + 0.05% Tween20. Sections stained with Ulex-biotin underwent a biotin-avidin block prior to primary staining (ab64212; Abcam, Cambridge, UK). After washing, secondary antibodies were added for 1 h in the dark in the same buffer. Sections were mounted with an aqueous mounting medium with DAPI (ab104139; Abcam, Cambridge, UK). For every sample, a no primary stain control was included.

Briefly, primary antibodies and lectins used in this study were: anti-blood group A antigen (1:150; mouse mAb IgM; HE-193; Invitrogen, Carlsbad, MA, USA); *Ulex europaeus I* lectin (1:50; anti-H; biotinylated; GTX01511; Genetex, Irvine, CA, United States); anti-C4d (1:100; rabbit mAb IgG; A24-T; ab136921; Abcam, Cambridge, UK); anti-C1qA (1:200; recombinant rabbit mAb IgG; EPR14634; ab189922; Abcam, Cambridge, UK); and, anti-C5b-9 (1:200; rabbit pAb IgG; A227; Complement Technology Inc., Tyler, TX, USA).

The secondary antibodies used were: anti-mouse IgM-AF555 (1:500; goat pAb IgG; A-21426; Invitrogen, Carlsbad, MA, USA); anti-rabbit IgG-AF488 (1:750; goat pAb IgG; A28175; Invitrogen, Carlsbad, MA, USA); and, streptavidin-AF647 (1:2000; S32357; Invitrogen, Carlsbad, MA, USA).

Slides were imaged using a Leica TCS SP5 confocal microscope (Leica Microsystems, Wetzlar, Germany) and six random fields of view were imaged per section.

### Image analysis
All images were analysed with FIJI software (v2.9.0)[47], with further details summarised in the Supplementary Information. Briefly, for quantification of antigen loss, a threshold of pixel intensity was determined on the anti-A channel based on the six images of the pre-treatment biopsy per experiment and the integrated density was measured per image. Per biopsy, six randomly selected fields of view were measured and normalised to the average value of the pre-treatment biopsy per experiment and all data points per cohort were collated (n = 3 kidneys per cohort, so 18 values per cohort). A similar procedure was used for *Ulex europaeus* (anti-H), and IgM-stained images.

### Flow cytometry assay for circulating antibody levels
To measure circulating antibody levels in the perfusate, 50 µl of 1% fixed type A RBCs were pipetted in a 96-well U-bottomed plate. Staining buffer (PBS + 0.6% BSA) or perfusate samples were added to the RBCs and incubated for 30 min at RT while shaking on a microplate shaker. A positive control was included in each plate using anti-A IgM (Lorne Laboratories Ltd, Reading, UK) diluted in PBS to a final titre of 1:128. RBCs were washed three times in staining buffer before resuspension in staining buffer containing goat anti-mouse IgM-647 (1:2000; A-21238; Invitrogen, Carlsbad, CA, USA) and were incubated for 30 min at RT in the dark while shaking. The unstained control received only staining buffer. RBCs were washed as before and resuspended in staining buffer and kept in the dark before flow cytometry.

For flow cytometry, the 96-well plate was analysed using an NXT Attune with Autosampler. RBCs and singlets were gated, and at least 10,000 events were recorded per sample (Supplementary Table 1 and Supplementary Fig. 7). Subsequent flow cytometry analysis was completed using FlowJo™ v10.9.0 Software (BD Life Sciences). Further information is provided in the Supplementary Information.

### NGAL measurements
Perfusate levels of NGAL were measured using a LEGENDplex™ assay from Biolegend (Kidney Panel 2 - 740583; San Diego, CA, USA) using a dilution of 1:150 for samples. The assay was completed as per the manufacturer's instructions using an NXT Attune with Autosampler and the data were analysed with LEGENDplex™ Data Analysis Software Suite (Biolegend, San Diego, CA, USA).

### C5a and sC5b-9 measurements
Perfusate levels of C5a (DY2037; R&D Systems, MN, USA) and sC5b-9 (558315; BD Biosciences, NJ, USA) were measured with ELISAs following the manufacturers' recommendations and using a FLUOstar Optima plate reader (BMG Labtech, Ortenberg, Germany).

### Statistical analysis
Grouped data at a specified timepoint were analysed with a non-parametric two-tailed Wilcoxon matched-pairs signed rank test with Holm-Sidak multiple comparisons correction, where appropriate, after normality testing with a Shapiro–Wilk test. Results were considered significant below $p = 0.0500$. Unless otherwise stated, average values at a specified timepoint are presented as mean ± standard deviation (SD). 95% confidence intervals (95% CI) are reported as [lower CI, upper CI]. All statistical tests were completed with GraphPad PRISM v10.1.1 (San Diego, CA, USA).

### Reporting summary
Further information on research design is available in the Nature Portfolio Reporting Summary linked to this article.

## Data availability
All data needed to reproduce this study can be found in the paper and Supplementary Information. Source data are provided with this paper.

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

## Acknowledgements

*Fp*GalNAc DeAc and *Fp*GalNase were provided for this study by Avivo Biomedical Inc. (Vancouver, B.C., Canada). This study was funded by the Stoneygate Trust (G114280) and the National Institute for Health and Care Research (NIHR) Blood and Transplant Research Unit in Organ Donation and Transplantation (NIHR203332), a partnership between NHS Blood and Transplant, University of Cambridge and Newcastle University. The views expressed are those of the author(s) and not necessarily those of the NIHR, NHS Blood and Transplant or the Department of Health and Social Care. S.MacMillan was funded by Kidney Research UK (G113265). This report is independent research. NHS Blood & Transplant have provided Relevant Material in support of the research. We would like to thank the renal pathologist Dr. Sathia Thirunavukkarasu.

## Author contributions

S.MacMillan. designed, completed, and analysed experiments, and wrote the manuscript for this study with feedback from all authors. S.H. helped with perfusion experiments and supervised the project. L.W.P. helped with perfusion and staining experiments. P.R., J.N.K.,

S.Macdonald., and S.G.W. produced the *Fp*GalNAc DeAc and *Fp*Gal-Nase for this study. M.L.N. supervised the complete study.

## Competing interests

S.G.W., J.N.K., S.Macdonald and P.R. are founders of Avivo Biomedical Inc., which is commercializing the enzymes described. J.N.K., P.R., and S.G.W. are inventors on patent application (PCT no. WO2020034043A1) submitted by the University of British Columbia. The remaining authors declare no competing interests.
