## [Peer Review File · Nature Communications]

Enzymatic conversion of human blood group A kidneys to universal blood group OREVIEWER COMMENTS

Reviewer #1 (Remarks to the Author):

This is a very well written manuscript on a relatively small study in pairs of discarded human donor kidneys, utilizing two enzymes to convert kidneys to a universal blood type O, through ex vivo hypothermic or normothermic machine perfusion.

I found this study to be timely and important, it addresses an urgent clinical need and presents a highly innovative intervention. It is quite surprising that HMP seems to be equally good as NMP at converting the blood type ex vivo, which will make it even more feasible (and economic) to translate these findings to clinical transplant cases.

I particularly liked the extensive and honest way in which this study's limitations are discussed.

My main concern is that, for NMP, the authors have used an acellular perfusate, i.e. without the addition of an oxygen carrier (such as red blood cells). It is rather established that a kidney, perfused ex vivo, does require an oxygen carrier in the perfusate at 37 degrees Celsius, for optimal oxidative metabolism. Why did the authors not add an oxygen carrier? Or, why did they not consider utilizing sub-normothermic perfusion (e.g. at 32 degrees) instead, to decrease the oxygen demand of the kidney ex vivo?

Reviewer #2 (Remarks to the Author):

In this work, MacMillan and co-workers successfully removed blood group A antigens from human kidney vasculature by using FpGalNAc DeAc and FpGalNase, from the bacterium *Flavonifractor plautii*. Specifically, using normothermic machine perfusion (NMP) and hypothermic machine perfusion (HMP) strategies, the authors demonstrated blood group A antigen loss of approximately 80% in 2hrs and 6hrs, respectively. In addition, they showed that treated kidneys do not bind circulating anti-A antibodies in an ex vivo model of ABO-incompatible transplantation and do not activate the classical complement pathway. This study represents the first report of blood group A removal in human kidneys, and the first

report of enzymatic conversion using hypothermic machine perfusion. This approach might increase utility of donor organs already retrieved for transplantation to improve overall waiting times for kidney transplantation. The content of the manuscript advances the field and it is appropriate for publication in the journal.

I have one comment that is non-essential but might be of some use. I suggest incorporating a Figure into the main text, showing/summarizing the histology of kidneys after NMP, HMP and ABOi perfusion, at key time points 2hrs, 6hrs, and 4hrs, respectively.

Congratulations to the authors for this very nice work.

Reviewer #3 (Remarks to the Author):

Macmillan et al report an interesting study on the conversion of blood type A donor kidneys to blood type O kidneys. This group has presented similar data on the conversion of blood type B kidneys (Macmillan Br J Surg. 2023 Jan 10;110(2):133-137), but the present study provides more details and is of greater relevance considering the larger pool of blood type A kidneys. The data show that conversion of A to O type can be achieved both at 37C, but also at 4C, albeit less efficiently. There were no signs of toxicity or loss of function during the study period. What is novel in this study is the exposure of A-stripped kidneys to anti-A antibodies, showing a proof-of-concept for avoiding hyperacute rejection in ABOi transplantation.

This is a well performed study. I have a few questions.

Why are the data in Fig 2B normalized? Non-normalised data and inclusion of T0 A and H antigen levels would provide better insight into the timing of antigen expression, and variation between kidneys.

After 10h HMP A antigen re-appears (Fig 3B), however there is no decrease in H antigen. Why does A return at 4C after 10h? And why is the return of A not accompanied by a decrease in H?

Why was chosen to test antibody binding to blood type converted kidneys using mouse monoclonal anti-A and -B IgM antibody and not blood type B or O serum? This would closer resemble clinical practice.

Table 2 shows quite a difference between kidneys in A antigen clearing. Can binding of anti-A antibody be correlated with clearing efficiency?

Did the authors try to see what happens after addition of coagulants by any chance in a pilot?

The authors describe the addition of anti-A and anti-B antibody to ABOe kidneys where anti-B is used as a control I suppose, however only data of anti-A is shown. Can the authors clarify this part of the study?

For the discussion on blood group A subtypes and differences in antigen level, is it known whether there a difference in clearing efficiency between AA and AO kidneys, as antigen levels are likely to differ?

Minor comments:

Introduction line 58-59 'Strategies to mitigate this wide gap in access to compatible transplants have been limited, with ABO-incompatible (ABOi) transplants providing one of the few clinically viable option.' Different allocation rules may also be a way to solve the problem?

Introduction; reasoning for the use of HMP over NMP for blood group conversion may also include the bacterial origin of the enzymes, which may not require 37C for optimal functioning?

Methods 3.1: the 9 kidney donors were all blood type A, right?

Please check grammar of 4.5 ABOe kidneys do not binding circulating antibodies in a reperfusion model of ABO-incompatibility

Point-by-point response to reviewer comments

We would like to thank all reviewers for their consideration of this manuscript and for their valuable feedback and comments. We would like to address the following points raised by the reviewers:

Reviewer 1:

This is a very well written manuscript on a relatively small study in pairs of discarded human donor kidneys, utilizing two enzymes to convert kidneys to a universal blood type O, through ex vivo hypothermic or normothermic machine perfusion.

I found this study to be timely and important, it addresses an urgent clinical need and presents a highly innovative intervention. It is quite surprising that HMP seems to be equally good as NMP at converting the blood type ex vivo, which will make it even more feasible (and economic) to translate these finding to clinical transplant cases.

I particularly liked the extensive and honest way in which this study's limitations are discussed.

My main concern is that, for NMP, the authors have used an acellular perfusate, i.e. without the addition of an oxygen carrier (such as red blood cells). It is rather established that a kidney, perfused ex vivo, does require an oxygen carrier in the perfusate at 37 degrees Celsius, for optimal oxidative metabolism. Why did the authors not add an oxygen carrier? Or, why did they not consider utilizing sub-normothermic perfusion (e.g. at 32 degrees) instead, to decrease the oxygen demand of the kidney ex vivo?

A consideration in the design of this study is chronically limited supply of blood group O red blood cells (RBCs) in blood banks nationally. Where in other perfusion experiments ABO-matched RBCs from blood group A or B donors can be used, the current investigation could not use blood group A RBCs as this would vastly increase the number of blood group A antigen substrates for *FpGalNAc* deacetylase and *FpGalactosaminidase*. This would be a major confounding variable in this study and thus a more reproducible acellular solution was chosen for normothermic machine perfusion (NMP). The acellular perfusate composition is chemically defined and not dependent on variable donor RBCs and preliminary data from our lab indicates that acellular perfusion conditions during NMP can maintain kidney function in a manner comparable to conditions using RBCs. NMP using an acellular solution has also been described by Thomas Minor's group in Essen, Germany, who used an acellular solution (Steen solution) to perfuse human kidneys before transplantation¹. Although the protocol used a rewarming strategy, kidneys were perfused at 35°C and then successfully transplanted.

Sub-normothermic machine perfusion was not considered for this study as *FpGalNAc* deacetylase and *FpGalactosaminidase* had previously been optimised for RBC blood group conversion at 37°C and had been validated during NMP of whole lungs previously^{2,3}. In addition, clinical use of NMP has only taken place at normothermic temperatures, thus perfusion at 37°C was deemed most translational for clinical application.

Reviewer 2:

In this work, MacMillan and co-workers successfully removed blood group A antigens from human kidney vasculature by using FpGalNAc DeAc and FpGalNase, from the bacterium *Flavonifractor plautii*. Specifically, using normothermic machine perfusion (NMP) and hypothermic machine perfusion (HMP) strategies, the authors demonstrated blood group A antigen loss of approximately 80% in 2hrs and 6hrs, respectively. In addition, they showed that treated kidneys do not bind circulating anti-A antibodies in an ex vivo model of ABO-incompatible transplantation and do not activate the classical complement pathway. This study represents the first report of blood group A removal in human kidneys, and the first report of enzymatic conversion using hypothermic machine perfusion. This approach might increase utility of donor organs already retrieved for transplantation to improve overall waiting times for kidney transplantation. The content of the manuscript advances the field and it is appropriate for publication in the journal.

I have one comment that is non-essential but might be of some use. I suggest incorporating a Figure into the main text, showing/summarizing the histology of kidneys after NMP, HMP and ABOi perfusion, at key time points 2hrs, 6hrs, and 4hrs, respectively.

We thank Reviewer 2 for this suggestion but due to our article including the maximum number of tables and figures permitted for an original article (10 total), we have been unable to provide this as a figure in the main text. We would like to highlight that the H&E histology for all conditions is provided in the supplementary for NMP (Supplementary Fig. 2), HMP (Supplementary Fig. 4), and HMP-ABOi (Supplementary Fig. 6b) cohorts and can be re-arranged as per editor preference.

Reviewer 3:

Macmillan et al report an interesting study on the conversion of blood type A donor kidneys to blood type O kidneys. This group has presented similar data on the conversion of blood type B kidneys (Macmillan *Br J Surg.* 2023 Jan 10;110(2):133-137), but the present study provides more details and is of greater relevance considering the larger pool of blood type A kidneys. The data show that conversion of A to O type can be achieved both at 37C, but also at 4C, albeit less efficiently. There were no signs of toxicity or loss of function during the study period. What is novel in this study is the exposure of A-stripped kidneys to anti-A antibodies, showing a proof-of-concept for avoiding hyperacute rejection in ABOi transplantation.

This is a well performed study. I have a few questions.

- a. Why are the data in Fig 2B normalized? Non-normalised data and inclusion of T0 A and H antigen levels would provide better insight into the timing of antigen expression, and variation between kidneys.

The decision to normalise the data in Fig. 2B to the level pre-treatment was to succinctly summarise the antigen expression data from all three kidney pairs in a meaningful manner. The raw data from this analysis is the measurement of the integrated density (the product of

fluorescence area and brightness) which is a value with arbitrary units that varies with each kidney donor and staining experiment. It would therefore be inappropriate to show a summary figure of integrated density values from pooled data as presenting this raw data is more suited to separate graphs per kidney pair. While this would provide a better insight into variation between kidneys, as Reviewer 3 has highlighted, we believe that showing separate graphs may be cumbersome and not necessarily translate to a clearer understanding of antigen expression changes for the reader.

Further to this, the inclusion of the raw data values of A and H antigen expression at time 0hr would be to show the baseline expression of H and A antigens. As the assumption here is that antigen expression is stable in the absence of treatment, we believe that the presentation of normalised data is appropriate, where the mean integrated density values at time 0hr are considered to reflect 100% expression.

- b. After 10h HMP A antigen re-appears (Fig 3B), however there is no decrease in H antigen. Why does A return at 4C after 10h? And why is the return of A not accompanied by a decrease in H?

The increase in A antigen expression shown in Fig. 3b is also described in more detail in Table 2 (and Table 3 for the H antigen). While the mean percentage of A antigen expression reaches a low of 13% at 5hrs, the increase to 29% antigen expression at 12hrs is not significantly different, with overlapping 95% confidence intervals. This also explains why H antigen levels do not appear to decrease during this period.

- c. Why was chosen to test antibody binding to blood type converted kidneys using mouse monoclonal anti-A and -B IgM antibody and not blood type B or O serum? This would closer resemble clinical practice.

The decision to use mouse monoclonal anti blood group antibody over human serum was to achieve a more reproducible perfusate composition, where antibody titre and immunoglobulin class were identical between experiments. Additionally, any serum added to the circuit would be diluted by the remaining perfusate composition (as outlined in the Supplementary Methods). This means that to achieve a clinically relevant titre of ABO antibodies in the perfusate (in this case, 1:128) with the dilution effect, the antibody titre in the human serum would need to be a minimum of 1:512 (assuming serum would replace maximally $\frac{1}{4}$ of the perfusate volume). It is not possible to obtain ABO antibody titres before obtaining and thawing serum, and only a small percentage of individuals have native ABO titres of above 1:128⁴. This made the addition of human type O or B serum in whole human kidney perfusion logistically non-feasible.

- d. Table 2 shows quite a difference between kidneys in A antigen clearing. Can binding of anti-A antibody be correlated with clearing efficiency?

This is an interesting point which is perhaps better approached during longer perfusion experiments as it appears that antibody binding is not necessarily saturated at 4hrs reperfusion and therefore may not reflect the extent of antigen clearing (Fig. 6d).

- e. Did the authors try to see what happens after addition of coagulants by any chance in a pilot?

No pilot data concerning the addition of coagulants to perfusion has been obtained to date, although this is in development in current studies. As suitable numbers of blood group specific human kidneys declined for transplantation and offered for research are difficult to obtain, the decision was made not to optimise the addition of coagulants prior to this study due to the concerns raised in the discussion.

- f. The authors describe the addition of anti-A and anti-B antibody to ABOe kidneys where anti-B is used as a control I suppose, however only data of anti-A is shown. Can the authors clarify this part of the study?

The addition of antibodies against both antigens was to recapitulate a model of A to O transplantation, where the modelled blood group O individual would have circulating antibodies against A and B. Only antibodies against the A antigen were able to bind native A antigens in the kidneys examined thus only these antibodies were experimentally examined in this study.

- g. For the discussion on blood group A subtypes and differences in antigen level, is it known whether there a difference in clearing efficiency between AA and AO kidneys, as antigen levels are likely to differ?

This is a very interesting question which we would like to investigate further. Very little research has examined ABO blood group antigen expression in human tissue with reference to donor genotype and the effect of enzyme treatment in this context is therefore unknown. As individuals with AA genotypes express more A antigens on RBCs compared to AO individuals⁵, it is sensible to hypothesise that a homozygous genotype may increase enzyme clearing time, but this may be mitigated by increasing dosage or optimising temperature conditions (using NMP vs HMP, for example). We aim to integrate ABO-allele specific genotyping into further work in this field and a note concerning this point has been added to the discussion (pages 22, lines 3-6).

Minor comments

- h. Introduction line 58-59 'Strategies to mitigate this wide gap in access to compatible transplants have been limited, with ABO-incompatible (ABOi) transplants providing one of the few clinically viable option.' Different allocation rules may also be a way to solve the problem?

This is a very valid point which has now been addressed fully in the discussion (page 22, lines 8-14) and the phrasing has been amended in the introduction (page 3, line 16).

- i. Introduction; reasoning for the use of HMP over NMP for blood group conversion may also include the bacterial origin of the enzymes, which may not require 37C for optimal functioning?

This is an interesting point, although the origin of these enzymes from bacteria found in gut flora and a previous study of successful whole lung conversion at 37°C supports the initial use of *FpGalNAc* deacetylase and *FpGalactosaminidase* in normothermic conditions^{2,3}. However, as these enzymes have never been tested during hypothermic organ preservation,

our study was designed to examine enzyme efficiency in both conditions in whole kidney conversion.

j. Methods 3.1: the 9 kidney donors were all blood type A, right?

All kidneys recruited to this study were blood group A. Methods 3.1 has been amended to state this (page 5, line 10)

k. Please check grammar of 4.5 ABOe kidneys do not binding circulating antibodies in a reperfusion model of ABO-incompatibility.

The results section 4.5 has been edited for grammatical errors and clarity (page 16, line 5).

References

1. Zlatev, H., von Horn, C., Kathes, M., Paul, A. & Minor, T. Clinical use of controlled oxygenated rewarming of kidney grafts prior to transplantation by ex vivo machine perfusion. A pilot study. *Eur J Clin Invest* **52**, e13691 (2022).
2. Rahfeld, P. *et al.* An enzymatic pathway in the human gut microbiome that converts A to universal O type blood. *Nat Microbiol* **4**, 1475–1485 (2019).
3. Wang, A. *et al.* Ex vivo enzymatic treatment converts blood type A donor lungs into universal blood type lungs. *Science Translational Medicine* **14**, eabm7190.
4. de França, N. D. G., Poli, M. C. C., Ramos, P. G. de A., Borsoi, C. S. da R. & Colella, R. Titers of ABO antibodies in group O blood donors. *Rev. Bras. Hematol. Hemoter.* **33**, 259–262 (2011).
5. Sharon, R. & Fibach, E. Quantitative flow cytometric analysis of ABO red cell antigens. *Cytometry* **12**, 545–549 (1991).

REVIEWERS' COMMENTS

Reviewer #1 (Remarks to the Author):

My feedback has been adequately addressed. I have no further comments.

Reviewer #2 (Remarks to the Author):

The authors have addressed all my comments.

Reviewer #3 (Remarks to the Author):

I thank the authors for the answers to my questions. Their answers were clear and rational, and a few additions were made to the manuscript to address the raised issues. I find this a very relevant manuscript and an important contribution to the field of organ transplantation.